# The Role of Risk or Contributory Death Factors in Methadone-Related Fatalities: A Review and Pooled Analysis

**DOI:** 10.3390/metabo11030189

**Published:** 2021-03-22

**Authors:** Arianna Giorgetti, Jennifer Pascali, Massimo Montisci, Irene Amico, Barbara Bonvicini, Paolo Fais, Alessia Viero, Raffaele Giorgetti, Giovanni Cecchetto, Guido Viel

**Affiliations:** 1DIMEC, Department of Medical and Surgical Sciences, University of Bologna, 40126 Bologna, Italy; arianna.giorgetti@unibo.it (A.G.); paolo.fais@unibo.it (P.F.); 2Department of Cardiac, Thoracic, Vascular Sciences and Public Health, University of Padova, 35121 Padova, Italy; jennifer.pascali@unipd.it (J.P.); massimo.montisci@unipd.it (M.M.); irene.amico@studenti.unipd.it (I.A.); studio.bbonvicini@gmail.com (B.B.); alessia.viero@aopd.veneto.it (A.V.); giovanni.cecchetto.1@unipd.it (G.C.); 3Department of Excellence of Biomedical Sciences and Public Health, University “Politecnica delle Marche” of Ancona, via Conca 71, 60126 Ancona, Italy; r.giorgetti@staff.univpm.it

**Keywords:** methadone, drug-related death, post-mortem examination, forensic toxicology

## Abstract

Methadone-related deaths are characterized by a wide range of post-mortem blood concentrations, due to the high pharmacokinetic/dynamic inter-individual variability, the potential subjective tolerance state and to other risk factors or comorbidities, which might enhance methadone acute toxicity. In the present study, the association among pre-existing and external conditions and diseases and the resultant methadone death capacity have been investigated. Beside a systematic literature review, a retrospective case-control study was done, dividing cases in which methadone was the only cause of death (controls), and those with associated clinical-circumstantial (naive/non-tolerant state), pathological (pulmonary or cardiovascular diseases) or toxicological (other drugs detected) conditions. Methadone concentrations were compared between the two groups and the association with conditions/diseases was assessed by multiple linear and binomial logistic regressions. Literature cases were 139, in house 35, consisting of 22 controls and 152 cases with associated conditions/diseases. Mean methadone concentrations were 2122 ng/mL and 715 ng/mL in controls and cases respectively, with a statistically significant difference (*p* < 0.05). Lower methadone concentrations (by 24, 19 and 33% respectively) were detected in association with naive/non-tolerant state, pulmonary diseases and presence of other drugs, and low levels of methadone (<600 ng/mL) might lead to death in the presence of the above conditions/diseases.

## 1. Introduction

The use of illicit opioids and the abuse/misuse of prescription opioids represent a global pandemic and a public health challenge [1]. While in the United States highly potent synthetic opioids, such as fentanyl and its derivatives, have been considered responsible for the so-called third-wave of an opioid crisis, most of the drug-induced deaths in Europe still involve traditional opioids such as heroin and methadone [1].

Methadone is a synthetic diphenylpropylamine which acts as an agonist on opioid receptors, especially on μ receptors, and also as a slight N-methyl-D-aspartate (NMDA) antagonist [2]. It is usually consumed by oral administration, rapidly absorbed with a bioavailability of 70–90% and extensively metabolized in the liver by cytochrome P450, mainly by N-demethylation and cyclization to 2-Ethylidene-1,5-dimethyl-3,3-diphenylpyrrolidine (EDDP) [2,3]. Due to its high bioavailability, long half-life (~24 h), low cost, and due to the availability of a specific antidote, methadone is widely used as a therapy for chronic pain and as a substitute for heroin in maintenance treatment [4,5]. The introduction of methadone maintenance treatments (MMT) in the case of heroin addiction has been shown to be effective in reducing drug-related crimes, diseases and deaths [6,7,8,9,10]. Methadone has been increasingly prescribed in the United States and in Europe, being used by 63% of patients in substitute treatment, while only 34% are in therapy with buprenorphine [1,11,12,13]. Methadone is generally recommended over buprenorphine due to its higher efficacy and lower costs [14], but the risk of fatal poisoning remains higher in MMT patients in comparison with buprenorphine [15,16,17]. Moreover, methadone is also commonly used for recreational purposes and it has been recently correlated to an upsurge of opioids deaths in France, Spain, Norway, Denmark and Sweden [1,18,19,20,21,22].

Methadone toxidrome typically includes miosis, dry mouth, sweating, anorexia, nausea, vomiting, hypotension, bradycardia and various grades of sedation from dizziness to somnolence until loss of consciousness and profound respiratory depression [23,24]. Central nervous system-mediated respiratory depression represents the main fatal hazard connected to methadone [2,4,24]. However, a cardiotoxic effect has also been described, affecting QT interval prolongation and arrhythmias, e.g., torsades de pointes (TdP) [25,26]. Thus, methadone should be prescribed with caution to patients at high risk for arrhythmias or for respiratory depression [27]. Studies involving fatalities in patients submitted to MMT have found that death is mostly related to medical illness (24%) or to a mixed drug toxicity [28]; additionally, the first weeks following the initiation or cessation of MMT are characterized by a higher risk of fatality, due to a decreased tolerance to methadone effects [3]. However, the impact of a reduced tolerance in methadone-related fatalities is not easy to be evaluated and has also been considered non-critical by some authors [29].

In forensic pathology, methadone could be an incidental finding during post-mortem examination or could be mainly responsible for death [30]. Due to high inter-individual pharmacokinetic and dynamic variability, the toxic effects of methadone can emerge at different doses [31], according to literature starting from 200, 400 or 600 ng/mL [32]. Interestingly, methadone has been also implicated in deaths at sub-toxic or therapeutic concentrations [33,34], making it hard to define a toxic threshold above which fatalities certainly occur. Indeed, therapeutic and fatal concentrations widely overlap in literature reports [19,24,35,36,37,38,39]; for example, the same methadone concentrations can be found in methadone-related deaths and in fatalities classified as “other causes of death” [19]. It is by far supposed and investigated through the literature that other contributory factors such as additional substances, opioid tolerance and cardio-pulmonary diseases might have an impact on fatalities in case of methadone assumption. However, the knowledge regarding the influence of such factors is still unknown and reviews or systematic studies on the topic are lacking.

In this challenging scenario, the aim of the present study is to provide a contribution to the issue, bringing further statistical and epidemiological evidence regarding the causal and concausal role of pre-existing intrinsic conditions, such as methadone non-tolerance and cardio-pulmonary diseases, or external conditions, such as co-assumption of other psychotropic substances. In particular, a review of selected literature beside the statistical evaluation of the casuistry of the Institutes of Legal Medicine of the Universities of Padova, Bologna and Ancona (post-mortem examinations performed according to the Recommendation R(99)3 of the Council of Europe on the Harmonization of Medico Legal Autopsy Rules [40,41,42]) was achieved and the following hypotheses (H) were considered:

**Hypothesis** **1** **(H1).**
*Lower methadone blood concentration would be found in the presence of clinic-circumstantial, toxicological and pathological conditions and diseases.*


**Hypothesis** **2** **(H2).**
*Methadone post-mortem lethal levels would show a relationship with the presence/absence of existing or external conditions.*


**Hypothesis** **3** **(H3).**
*Low concentrations of methadone would be expected in victims bearing several co-occurring conditions and diseases.*


## 2. Results

### 2.1. Literature Review and Cases

The literature review resulted in 207 articles, with a final inclusion of 84 papers (Appendix A) [16,19,20,29,34,35,36,37,38,39,43,44,45,46,47,48,49,50,51,52,53,54,55,56,57,58,59,60,61,62,63,64,65,66,67,68,69,70,71,72,73,74,75,76,77,78,79,80,81,82,83,84,85,86,87,88,89,90,91,92,93,94,95,96,97,98,99,100,101,102,103,104,105,106,107,108,109,110,111,112,113,114]. Among the included papers, 66 were retrospective/prospective case series (79%), 11 case reports (13%), 3 original articles (3%), and 4 reviews (5%).

Twenty-eight papers matched criteria 4–5, and were, thus, included in the case selection process, providing 139 literature cases (Appendix A). Results of the selection process are shown in the Prisma flow chart (Appendix A).

Victims were mostly males (77% of cases), with a mean age of 33. Mean post mortem methadone blood concentration was 872 ng/mL. Twenty-one percent of victims were deemed naive/non-tolerant on the basis of the data reported in the corresponding papers.

Results of the post-mortem examination regarding lungs and heart were reported in 23 cases out of 135 and in 15 of these, pulmonary edema was described. The following pulmonary diseases were seen: bronchopneumonia (5 cases) chronic bronchitis (3 cases), emphysema (1 case), unspecified inflammation (2 cases). Cardiovascular diseases were represented by cardiac hypertrophy (8 cases), heart enlargement/dilatation (4 cases), endocardial or interstitial sclerosis (2 cases), myopericarditis (2 cases), lipomatous infiltration of the right ventricle (2 cases), others, including severe coronary artery disease or aortic coarctation (3 cases). Cardiovascular and pulmonary diseases co-occurred in 7 subjects. When no disease was reported, its absence was assumed.

At least one drug beside methadone was detected in 96 (69%) literature cases. Cocaine and BEG were described in 9 and 21 cases. Nineteen cases tested positive for morphine or codeine. BZDs were found in 47 cases, 37 of which had diazepam, 33 nordiazepam, 10 oxazepam, 5 temazepam, 3 bromazepam and lorazepam both, 2 delorazepam and 7-aminoflunitrapeam both, 1 7-aminoclonazepam and 1 desalkylflurazepam. Other drugs were seen in 64 cases. Particularly, ethanol was present in 26 cases, while antidepressants, including amitriptyline, citalopram, doxepin, mirtazapine, nortrimipramine, nortriptyline, norsertraline, sertraline, trimipramine, venlafaxine, were found in 27 cases. THC and metabolites were detected in 16 cases; neuroleptics and antipsychotics such as carbamazepine, chlorprothixene, levomepromazine, promazine and quetiapine, were identified in 10 cases. Amphetamines/methamphetamines were found in 6 cases, while tramadol and buprenorphine in 2 cases each.

### 2.2. Literature Review and Cases

Thirty-five in-house cases were retrospectively observed between 2008 and 2020 (Table 1 and Appendix A). Males (74%) in the third decade (mean age 36) were the most represented methadone-related victims. Mean post-mortem methadone blood concentration was 934 ng/mL. Naive/non-tolerant subjects were 23%, as emerged by clinical-circumstantial data, which was always available. All cases displayed pulmonary edema, as assessed by weighing of the lungs and histology, and 16 cases additionally exhibited pulmonary diseases, particularly bronchopneumonia (8 cases), chronic inflammation (7 cases) and emphysema (1 case).

Among cardiovascular diseases, endocardial or interstitial sclerosis (11 cases) were the most represented. Heart enlargement/dilatation (2 cases), myopericarditis (2 cases) and septic emboli (1 case) were also described.

In 29 cases (83% of the casuistry) an additional drug, other than methadone, was detected. Cocaine or BEG were quantified in a total of 17 cases. Morphine was identified in 7 cases, among which 2 also tested positive for codeine and 1 for codeine and 6-MAM. Four cases also tested positive for BZDs. Ethanol was positive in 14 cases. Four cases presented antipsychotic/neuroleptics and 2 showed the presence of an antidepressant (Table 1).

Within the in-house casuistry, 4 cases (11%) had methadone as the only cause of death (CN), while 31 (89%) were classified as CA.

Detailed epidemiological data, blood methadone concentrations, conditions and diseases and CA/CN composition of literature and in-house cases are shown in Figure 1.

A summary of the drugs, beside methadone, detected in literature and in-house cases is shown in Figure 2.

### 2.3. Data Analysis

#### 2.3.1. Preliminary Analyses and H1 (Cases and Controls)

Literature and in-house pools did not show any statistically significant difference in epidemiological data, methadone blood concentration and CA/CN composition (*p* > 0.05). Thus, they were merged into a single database. The merged pool consisted of a total of 174 cases, among which 125 (72%) had other drugs beside methadone in post mortem blood, 27 (16%) had pulmonary diseases, 27 (16%) had cardiovascular diseases and 36 (21%) were naive/not tolerant to methadone.

Overall, 152 cases presented at least one pre-existing or external condition or disease and were thus considered as CA, while 22 cases were classified as CN. Mean methadone concentration was 715 ng/mL (SD 1112) in CA and 2122 ng/mL (SD 3644) in CN. A Mann-Whitney test resulted in a statistically significant difference in methadone levels between CA and CN (*p* < 0.001) (Figure 3).

#### 2.3.2. H2: Single and Multiple Linear Regression

Statistically significant linear regressions were found by plotting methadone blood concentrations with variables *d*–*f*, i.e., presence of other drugs, number of drugs and number of co-occurring conditions or diseases. Variables *a*–*c* did not show a linear relation with the dependent variable (*p* > 0.05).

When the dependent variable was transformed on a logarithmic base (base 10), all independent variables, except for variable *c* (cardiovascular diseases), resulted linearly related to blood methadone concentration. *p* values and equations are shown in Table 2.

The multiple linear regression model built by testing the dependent variable with variables a–d was statistically significant (*p* < 0.001), with R^2^ = 0.15. Table 3 reports the estimated factors of the independent variables on the log-transformed concentrations of methadone within the multiple regression model.

Coefficients allow to calculate the increase/decrease of blood methadone concentration for each unitary increase of the independent variables. Considering that the dependent variable was transformed on a logarithmic base, the corrected coefficients allowed to predict that naive/non-tolerant state, pulmonary diseases and the presence of other drugs in blood are associated with a reduction of the fatal blood methadone concentration by 23.86%, 19.12% and 32.71%, respectively.

Tolerance state and presence of other drugs weighted the most, while cardiovascular diseases were not significantly associated with lower lethal methadone concentrations.

There was no evidence of multicollinearity, with Pearson correlation always *r* < 0.5 (acceptance level *r* < 0.8). Collinearity was also checked and found at an acceptable level by coefficients analysis (tolerance 0.7–0.9; acceptable level >0.2).

#### 2.3.3. H3: Binomial Logistic Regression

Methadone post-mortem blood concentrations were recoded into “low” (<600 ng/mL) and “high” (>600 ng/mL). Based on the results of the multiple linear regression, a predictive model for a post-mortem detection of low and high methadone blood concentrations was built by binomial logistic regression model, firstly by testing variables *a*–*d*. The model was statistically significant, with a Nagelkerke R^2^ = 0.15 and a 68.4% correct classification. All variables, except variable *c* (cardiovascular diseases), were statistically significant predictors of low methadone blood concentrations, with positive odds ratio > 3 and *p* < 0.05 (Table 4).

A second binomial logistic regression model was then attempted with variables *e*–*f*, which were the only non-dichotomous and were linearly related to the non-log transformed methadone blood concentrations. Statistical significance predictors were also found in the number of drugs and in the number of co-occurring conditions or diseases, which had positive coefficients ratio > 1 (also reported in Table 4).

For every one-unit increase of the predictors, i.e., number of co-occurring conditions or diseases and number of drugs, there was an increasing likelihood of falling into the category of low detected concentrations of methadone (<600 ng/mL).

## 3. Discussion

The high number of papers found in the literature, including some very recent scientific productions, demonstrates that methadone-related death still represents an on-going issue in forensic medicine. In fact, due to the wide pharmacokinetic and dynamic variability, tolerance and post-mortem redistribution, the diagnosis of a methadone-related death is still challenging for the forensic pathologist [34]. Circumstantial data, underlying diseases, co-assumed drugs and reduced tolerance are important factors to be considered when a diagnosis of methadone fatal intoxication is under examination. This topic has been evaluated by means of several valuable retrospective case series, either specifically focused on methadone or considering and comparing more than one illicit drug or opioid dependence treatment, even though systematic review, meta-analysis and experimental studies were lacking.

The literature so far produced has suggested that victims of methadone-related death are mostly represented by males over 30 years of age [19,20,36,44,68,72,73,83,87,88,91,92,94,95,97,98,99,101]. The epidemiology of victims of the in-house casuistry was consistent with that of literature cases. This suggested that methadone overdoses might involve not only young and unexperienced users and that a long history of drug use/abuse does not totally exclude a certain risk of death [115].

In the pooled database, involving both literature and in-house cases, victims bearing pre-existing conditions or diseases showed lower methadone blood concentrations than those in which methadone only was detected, so far confirming the first Hypothesis (H1). Statistical models were applied to assess the relationship between single conditions/diseases, which are discussed below, and detected lethal methadone blood concentrations (H2). Moreover, an increasing number of conditions/diseases and of co-consumed drugs were predictors of the detection of low (<600 ng/mL) methadone levels (H3).

### 3.1. Tolerance to Methadone

The issue of tolerance to methadone is one of the most complex. In the literature, it has been described that fatal respiratory depression might occur at doses of 20 mg in non-tolerant subjects. This could be due to an accumulation of methadone in tissues, so that a rapid increase in the prescribed dosage might suddenly lead to unexpected high blood levels [36]. However, both the exact dose consumed by subjects and the tolerance state are difficult to be assessed during a post-mortem examination. The term “non-tolerant” includes naive subjects as well as those who have lost tolerance, e.g., as a consequence of an abstinence period [36] and it has been shown that when tolerance is excluded, even low blood concentrations of methadone can be fatal [73].

Our literature review has showed that deaths due to methadone occurred mostly in the first week after the beginning of MMT or in subjects who are not participating to MMT [24,36,48,49,77,91,112,116,117]. In a study by Bernard et al., a significant difference in methadone levels was encountered in the comparison between MMT and non-MMT victims and this difference was seen across all type of fatalities, from those involving methadone as the sole cause to those involving additional drugs at non-lethal and lethal levels [113].

As shown by this study covering multiple linear regression, naive/non-tolerant state was associated with lower blood methadone concentrations. Particularly, detected lethal post-mortem concentrations were 24% lower in victims showing a naive/non-tolerant state. Several retrospective case series have also shown that victims who were not under MMT at the time of death had significantly lower methadone concentrations than those in actual treatment (360–400 ng/mL vs. approx. 1000 ng/mL), suggesting a high tolerance in those who had a daily intake [19,24,35]. However, this was not consistently described in the literature [36]. Abstinence was demonstrated as a non-critical factor for heroin overdose deaths [118] and, similarly, no evidence that a reduced tolerance was the main cause of methadone-related death has been found [29].

Singularly considered, the enrollment of a subject in an MMT program does not necessarily reflect a tolerance to respiratory depressant effects, which may require a longer time to be settled compared to other methadone effects [119]. Full tolerance to narcotic effects can take up to 12 months in some individuals [38], while, on the contrary, recreational methadone users might display significant tolerance [96]. Thus, caution is generally needed in the estimation of tolerance acquiring both clinical and toxicological data when possible. Moreover, it has to be remembered that even experienced users are at risk of fatal intoxication [43,113], as confirmed by the high prevalence of patients enrolled in MMT for long time [20,43,112].

### 3.2. Respiratory Diseases

Methadone depresses the central nervous system respiration control center and affects respiration by firstly reducing the volume of inhaled and exhaled air and then the rate of respiration [24,38]. Cardiopulmonary failure, secondary to central nervous system depression, has been shown as the most important mechanism of death [92] and is consistent with the finding of lung edema in the majority of cases of our pooled database [60,83,101]. Particularly, lung edema was described in each case of the in-house casuistry. Since methadone depresses the cough reflex, an aspiration of gastric content and a secondary pneumonia might develop, causing upper- and lower-respiratory tract infections, common complications of opiate abuse. Underlying diseases altering the respiratory function, such as bronchopneumonia, emphysema, bronchitis, fibrosis and asthma, are likely to compromise respiratory capacity in opioid users [24,36,51,116]. For example, they have been proven to have an impact on oxycodone concentrations [93] and have been suggested as factors to identify vulnerable methadone users [46].

In a comparative study between heroin and methadone users, methadone deaths more frequently displayed systemic diseases [51]. In Albion et al., it was deemed that methadone exacerbated the effect of a pre-existing pneumonia in 5 cases, leading to death [43]. In the study of Milroy et al., bronchopneumonia was statistically more frequent in deaths ascribed to methadone only, rather than in those due to mixed drug toxicity. This could point towards a higher risk of death in the presence of pulmonary comorbidities, but the high prevalence of pulmonary infections was interpreted as a sign of a slow death with subsequent pulmonary inhalation [34]. No difference in methadone median concentration was found in a recent study between victims displaying mild to severe histopathological findings of bronchopneumonia [91]. Similarly, the effect was deemed non-relevant for methadone concentration in a recent multivariate analysis [93]. In a retrospective case series by Bernard et al., methadone concentrations in MMT patients suffering from somatic diseases were not significantly different from those with no somatic diseases [113].

In this study, pulmonary diseases were evaluated and, as shown by multiple regression analysis, were found to be associated with lower methadone concentrations, suggesting that pulmonary pre-existing comorbidities might act as risk factors for methadone-related death, though this was the least important among the conditions/diseases considered. Particularly, pulmonary diseases were associated with a decrease in the detected fatal methadone blood levels of about 19%. According to these data, pulmonary diseases in a death related to methadone should be considered with caution, keeping in mind that they might lead to death even at lower methadone concentrations, especially in combination with other risk factors.

### 3.3. Cardiovascular Diseases

Methadone can alter the membrane homeostasis and facilitate arrhythmias and sudden cardiovascular collapse [24,43]. Systemic comorbidities are quite common in opioid intoxications [120] and methadone-related intoxication victims are more likely to display cardiac diseases compared to heroin intoxications [51]. Coronary artery atherosclerosis, myocardial necrosis, cardiac enlargement and many other cardiovascular findings were described in 40 out of 206 cases reported by Pilgrim et al. [36] Mijatovic et al. have shown heart pathological changes in methadone deaths in accordance with the duration of addiction [60]. In this context, it would be reasonable to assume that additional arrhythmogenic factors, such as cardiac enlargement, myocardial necrosis and cardiac pathology, might facilitate TdP or just increase hypoxia in subjects under methadone.

Cardiomegaly was reported in association with methadone in the lower range of toxicity [43], but recent studies have not found different methadone concentration in subjects with and without cardiovascular diseases [93]. Sudden cardiac deaths have been associated with methadone even in the absence of significant cardiac abnormalities [33]. In a recent study, all methadone overdoses were associated with abnormal ECG findings, both in methadone-addicted and naive subjects, who did not show any myocardial and coronary artery diseases. This was interpreted as a clue for a high risk of ECG abnormalities in methadone overdoses and for a likely toxic effect on myocardium [121]. In a retrospective case series, sudden cardiac deaths in therapeutic methadone users were compared with deaths due to methadone intoxication in recreational users: sudden cardiac deaths in non-methadone users and histopathological macroscopic and microscopic cardiac findings did not differ between the tested groups [96].

According to these literature findings, the presence of cardiovascular diseases was not statistically associated with lower lethal methadone concentrations. In the in-house series, victims showing cardiovascular diseases, mostly myocardial sclerosis, had a concentration of methadone ranging from 22 ng/mL to 1450 ng/mL. Sudden cardiac death might occur even in the absence of underlying or pre-existing cardiovascular diseases.

Nonetheless, relevant cardiovascular abnormalities should never be under-estimated during post-mortem examinations, also taking into account that the presence of other risk factors (e.g., the consumption of other drugs or medicines, especially BZDs) might precipitate the cardiotoxic effect of methadone and further increase the risk for arrhythmias even at lower methadone concentrations [60,96,122].

### 3.4. Combined Drug Toxicity

As shown in several retrospective case series, the majority of methadone-related deaths displays a poly-drug consumption [19,20,24,35,36,37,44,52,60,80,84,85,88,92,94,109,111,112,113,116]. Though, the impact of co-consumed drugs on blood methadone concentration is inconsistent in the literature.

The median methadone level was found lower in mixed drug deaths with respect to methadone-only fatalities [34], and methadone concentrations significantly decreased when more than 3 drugs were involved [52], suggesting that polysubstance use might have a synergistic effect and could increase the risk of death. Deaths also occur at methadone concentrations <200 ng/mL when additional drugs are present, and particularly central nervous system depressants such as BZDs, alcohol and other opiates [30,34,35,83]. Mean concentrations were found lower in a methadone-diazepam mixed consumption with respect to methadone-only fatalities, and even lower methadone levels were found when other drugs were involved [60].

In other literature cases, similar methadone concentrations have been described in polydrug fatalities and in methadone mono-intoxications [43,52,62,111,114]. In a retrospective study, the median concentration in methadone-only drug and methadone in combination with other drugs was similar and slightly higher in the second group [91]. One possible explanation for this finding is represented by a competitive mechanism of different drugs on the same metabolic patterns, resulting in higher unmetabolized levels of methadone [43]. This does not seem to be confirmed by this study. Indeed, the number of other drugs detected resulted in statistical significance associated with the detection of lethal low methadone levels (<600 ng/mL) and, in multiple regression analysis, the co-consumption of other drugs represented the risk/contributory with the highest impact on log-transformed blood methadone concentrations.

In the majority of literature series and cases, BZDs were the most frequent co-consumed substance in addition to methadone [19,20,28,35,36,37,44,60,80,81,83,85,91,97,98,109,111]. BZDs were also the second most common drug detected in the merged database. BZDs are rarely seen as a standalone finding in post-mortem cases and can hardly produce a severe respiratory depression when singularly consumed. In the merged pool, BZDs were mostly found in therapeutic concentrations [32]. However, even these levels have been considered sufficient to induce a muscle relaxation and might be enough to contribute to death in combination with methadone [29,49]. The effect of BZDs was found unrelated to post-mortem concentration by Caplehorn et al. [49], while no statistically significant difference among methadone concentrations in the presence or absence of BZDs was described by Fields et al. [52] In our model, BZDs were not considered inferior to other drugs as possible risk/contributory factors and a significant impact of the variable was found. Though, further studies are needed in order to confirm this hypothesis.

The most abundant co-consumed drug in the in-house and merged database was cocaine, which was not as common as in the literature. However, cocaine use is highly prevalent among MMT patients [123] and cocaine was notably found in 1/3 of the cases described by the recent retrospective series [43,57]. Stimulants usually antagonize and minimize the respiratory depressant opioid effect; mechanisms by which cocaine might increase the probability of a methadone-related death are still to be comprehended and are beyond the aims of the present study.

Ethanol was also extremely common in the literature and in in-house revision. The impact of ethanol intake could be highlighted considering some cases, in which fatalities occurred at methadone concentrations of 160 ng/mL, in combination with a blood alcohol concentration (BAC) of 3.39 g/L [37] or still in the verge of toxicity (600 ng/mL) with BAC 1.1–2.29% [35]. These cases were also naive/non-tolerant to methadone effects, confirming a synergic effect of multiple drug intake and absence of tolerance. A reduction of predicted opioid concentration has been shown in fatalities in which ethanol was co-consumed, disregarding its levels [71,93]. A similar analysis has been attempted by, e.g., Bernard et al. [113], who showed decreasing concentrations in three groups represented by methadone-only victims, methadone combined with other drugs and fatalities mostly due to other drugs, although the difference did not reach a statistically significant level.

Even though clearly no distinction was applied among different types of drugs, our results support the hypothesis that the presence of other drugs in blood, especially in combination with other risk factors [115], might enhance methadone-related acute toxicity; indeed, in our model a 33% decrease of post-mortem blood concentration of methadone was observed.

### 3.5. Limitations

#### 3.5.1. Selection and Classification of Cases

Some limitations should be considered when approaching these data. The literature search was performed in a single database (PubMed). Even though the references of the included manuscripts were sought to check for further relevant studies, the present review cannot be classified as a systematic review of evidence. The selection process did not take into consideration the manner of death, which is strongly influenced by circumstantial data. Although non-accidental deaths might display higher methadone concentrations, the number of these cases is likely low with respect to the totality (<5%), thus the bias is considered not to invalidate the overall study. The time period for literature cases selection was limited to the last 20 years, which was considered a suitable period to give a modern vision of the problem.

We were aware that post-mortem redistribution might affect the reported methadone blood concentrations. In order to possibly limit this bias, in each case only peripheral blood concentrations sampled within a relatively short interval (3 to 5 days in in-house casuistry, exclusion of putrefied bodies) to death were considered. Though, a redistribution or a drug metabolism, e.g., due to a prolonged agony time, leading to partially lower methadone levels, cannot be excluded [34].

A further limitation of the study is represented by the classification of the tolerance state. For literature cases, since clinical-circumstantial information might have been missing, results of hair analyses were also considered in the evaluation of the tolerance state. On the other hand, hair material was not always available for toxicological analyses within the in-house casuistry. As a consequence, naive/non-tolerant state was assumed only in cases in which it was certain, while tolerance was assumed when not otherwise specified or inferable. Similarly, the absence of cardio-pulmonary comorbidities was assumed when no data was specified in the literature and this might have partially biased the results. Efforts were made to be complete and accurate when entering relevant information from each case into the database. A further limitation might be seen in the fact that some of the included articles were inhomogeneous as for study design and statistical methods used. In order to reduce to the minimum this bias, only articles with extractable cases were selected and used for the pooled analysis.

Finally, pre-existing and external conditions and diseases were organized into broad categories, without a distinction between singular diseases or severity of the pathological state and not considering the concentration of the additional drugs beside methadone. This was motivated by the relatively low number of cases showing pulmonary and cardiovascular diseases and would deserve further studies and evaluations.

The pharmacogenetics of the individuals was not tested in the present study. Though the literature on this topic, regarding methadone, is still limited, this could be a further factor of inter-individual variability and future studies might involve also an in-depth pharmacogenetic analysis.

#### 3.5.2. Statistical Analyses

One main limitation for statistical analysis may arise from the difference in the number of controls (CN: *n*. 22) compared to cases with risk/contributory factors for methadone-related death (CA: *n*. 152). Unfortunately, notwithstanding the thorough research in literature and in in-house casuistry, fatalities concerning the detection of methadone alone are rare. In fact, for example, patients in MMT are usually supplied with antipsychotics and/or benzodiazepines, long-term methadone addicted frequently present abnormalities in both pulmonary or cardiological levels etc. The further and continuous collection of control cases will permit to verify and improve the proposed statistical model.

Additionally, some factors have not been included in the analysis, such as age of the victims, body mass index, route of administration, pharmacogenomic data, post-mortem interval. As per other licit/illicit drugs, no single concentration drug level was considered, but only the presence/absence of the molecules. Finally, for binary logistic regression, cut-off values reported by Schulz et al. were employed [32].

## 4. Material and Methods

### 4.1. Literature Review and Cases

A systematic literature review was performed in June 2020 by two authors (IA, GV) in a public electronic database (PubMed), regarding deaths involving methadone. The inquiry strategy was: “((methadone and (death and forensic))) OR ((“Methadone” [Mesh]) AND “Death” [Mesh])”. References of the included papers were additionally screened for possible inclusions.

Inclusion criteria were:1.Date of publication, i.e., articles published from 2000 to 2020;2.English language;3.Topic, i.e., deaths in which methadone was assessed as a primary or contributory cause of death by at least a toxicological analysis revealing methadone in blood and at least an external examination.

Titles and abstracts were screened for criteria 1–3 by two blinded independent investigators (IA, GV). Included studies and those in which criterion 3 could not be assessed by title/abstract were further assessed for eligibility by reading full-texts. In cases of discrepancies, a physical consensus was held until issue resolution. For each paper, authors, title, journal, type (i.e., case report: ≤2 cases; case series: ≥3 cases; retrospective case series: revision of a wide casuistry with applied statistics; original article: experimental study) and year of publication were extracted.

In order to build a database of literature cases, the articles included in the literature review were re-assessed for eligibility by applying the following inclusion criteria:

4.Quantitative toxicological analysis performed on peripheral blood samples (femoral, preferable, or iliac vein) by gas-chromatography or liquid-chromatography mass spectrometry (GC-MS or LC-MS) with validated methods;5.Extractable data for single cases.

Exclusion criteria were:

6.Absence of quantitative data on peripheral blood (e.g., urine or qualitative analysis);7.Moderate or severe body decomposition.

### 4.2. In-House Cases

A retrospective analysis of the forensic autopsy reports of the Institutes of Legal Medicine of Padova, Bologna and Ancona between January 2008 and June 2020 was conducted, by selecting methadone-related deaths (corresponding to criteria 3–4). The causes of death were based on a comprehensive evaluation of circumstantial, autopsy, histology and toxicological reports. In all cases, the post-mortem examination was performed according to the Recommendation R(99)3 of the Council of Europe on the Harmonization of Medico Legal Autopsy Rules [40,41,42]. Particularly, in each case the ascertainment included a thorough external examination of the corpse followed by a complete internal examination with the opening of all three body cavities, the weighing of all major organs, the collection of tissues and biological fluids, and particularly the sampling of peripheral (femoral) blood, as well as a histological analysis of the main organs (hematoxylin and eosin staining). According to exclusion criterion 7, in each case the post-mortem examination was performed 3 to 5 days from the death, and thus putrefied bodies were not considered for the study. Moreover, cases in which a cardio-pulmonary resuscitation took place were discarded.

Toxicological analyses included an immunoenzimatic (EMIT or CEDIA) screening on peripheral blood for drugs of abuse and other psychotropic drugs, followed in each case by GC-MS or LC-MS confirmation by using validated methods routinely applied for forensic purposes (exclusion criterion 6). Ethanol was also determined in blood by headspace GC coupled to a flame-ionization detector (FID).

### 4.3. Data Analysis

#### 4.3.1. Data and Sample

Literature and in-house cases were reviewed by two experienced forensic pathologists and toxicologists (RG, GC) and two separate databases were prepared with the following data.

8.Epidemiology: sex and age of the victim;9.Tolerance to methadone, as evaluated by a comprehensive analysis of all clinical-circumstantial available data, including age, circumstances of death, participation and duration of MMT. Since this is a complex topic, tolerance was assumed when not otherwise specified or inferable and a naive/non-tolerant state was assigned only when it was certain;10.Post-mortem examination: macroscopic and histopathological findings, and particularly pulmonary (e.g., asthma, chronic obstructive pulmonary disease, bronchitis, bronchopneumonia) and cardiovascular diseases (e.g., cardiomegaly, hypertrophy or dilatation, severe atherosclerotic coronary artery disease, myocarditis, myocardial infarction). As performed in a previous study, when no information regarding pulmonary and cardiovascular diseases was available in the literature, the absence of significant findings was assumed;11.Toxicological data: concentration of methadone detected in peripheral blood, and the presence of other substances in blood, particularly ethanol, cocaine and its metabolite benzoylecgonine (BEG), heroin and its metabolites morphine, 6-Monoacetylmorphine (6-MAM), codeine, benzodiazepines (BZDs) and other neuroleptics/antidepressants.

#### 4.3.2. Descriptive Statistics and Preliminary Analyses

Literature and in-house databases were evaluated by descriptive statistics and, on the basis of the extracted data, cases were classified into two categories:cases in which methadone was associated with other clinical-circumstantial (naive/non-tolerant state) or pathological (pulmonary or cardiovascular diseases) or toxicological (other drugs detected in blood) conditions (so-called cases, CA);cases in which methadone was the only cause of death (so-called controls, CN).

In order to check if a pooled database could be built containing both literature and in-house cases, epidemiological data, methadone blood concentrations and CA/CN compositions were compared by descriptive statistics and t-tests (Mann–Whitney for methadone concentrations), setting a *p* value < 0.05.

#### 4.3.3. Hypotheses and Experimental Setting

To test the first Hypothesis (H1), methadone concentrations of CN and CA were compared by Mann–Whitney test.

To test the second Hypothesis (H2), dependent and independent variables were tested by single and multiple linear regression analysis.

Finally, the third Hypothesis (H3) was assessed by binomial logistic regression, which allows us to calculate the odds ratio, i.e., the probability of falling into a category on the basis of predictors.

#### 4.3.4. Measures

The dependent variable for this study was methadone concentration as quantified on blood samples collected during forensic autopsies.

**H1.** For Mann–Whitney test, no modification of the dependent variable was applied.**H2.** For single and multiple linear regression, the variable was investigated with no transformation and by means of log-transformation, to satisfy the linear regression assumptions.**H3.** For binomial logistic regression, the dependent variable was dichotomized into 0 (“low”, i.e., <than 600 ng/mL) and 1 (“high”, i.e., >than 600 ng/mL). In the absence of consensus regarding toxic and fatal levels, this threshold was based on the recent publication of Schulz M et al. [32].

The following independent variables were evaluated:naive/non-tolerant state;presence of pulmonary diseases;presence of cardiovascular diseases;presence of other drugs in blood;number of drugs detected beside methadone. When both a parent drug and its main metabolites were detected, a single drug was considered;number of pre-existing and external conditions and diseases, among the following: naive/non-tolerant state, cardiovascular diseases, pulmonary diseases, other drugs detected in blood.

For variable *a*, 0 = tolerance to methadone, 1 = naive or non-tolerant state. Variables *b*–*d* were dichotomized as 0 = absence, 1 = presence. Variables *e*–*f* were ordinal, ranging from 1 to 4.

#### 4.3.5. Statistical Analyses

Statistical analysis was performed by GraphPad Prism 8.2.1. and by SPSS Statistics, IBM© 26.0.0.0. Findings were considered statistically significant when *p* < 0.05. For multiple linear regression, collinearity diagnostics was run to check for multicollinearity of data. Durbin Watson and normal probability plot were performed to check if residuals were independent and normally distributed. Cook’s Distance test was run to test for influential cases, potentially biasing the model. The multiple linear regression herein reported permit the estimation of the effect of the independent variable (*a*–*f*) on log-methadone concentrations. Though the method allows us to explore relationships among variables, it does not establish causation. The percentage difference on the original concentration scale for each factor was calculated using the formula: (eβ − 1) × 100 (where β is the linear regression coefficient associated with that factor) to present the findings in a clinically relevant manner.

The binomial logistic regression predicts the probability that the detected lethal post-mortem blood methadone concentration falls into the category of “low” or “high” for every one-unit increase of the predictors or independent variables. As a measure of the variation explained by the model, the Nagelkerke R^2^ (possible values: 0–1) was assessed.

## 5. Conclusions

The present study confirms the interest of the scientific community on the topic of methadone-involving fatalities and highlights the difficulties in the post-mortem interpretation of these deaths, due to several factors including pharmacokinetic and dynamic variability, tolerance and post-mortem redistribution. As a principle, the diagnosis of the causal or concausal role of a substance and of additional factors in a death should be based on multiple elements, including toxicological and histology results.

The presented data have provided additional evidence on the impact of pre-existing and external conditions and diseases on deaths involving methadone and, particularly, on fatal blood methadone concentrations.

After a review of the relevant literature and a pooled analysis between literature and in house cases, it was found that deaths involving only methadone (controls) had higher concentrations than those so-called cases that presented additional clinical-circumstantial, pathological or toxicological conditions or diseases. The derived statistical model, by multiple linear regression, allowed to predict that naive/non-tolerant state, pulmonary diseases and the presence of other drugs/medicine might be considered risk factors for methadone-related death, being associated with the of lower methadone concentrations detected post-mortem. Particularly, 24%, 19% and 33% lower methadone concentrations have been seen in association with naive/non-tolerant state, pulmonary diseases and presence of other drugs/medicine. Thus, our study supports the evidence that lower levels of methadone might be enough to lead to death in the presence of these conditions/diseases compared to cases in which methadone is detected alone. By increasing the number of conditions and diseases co-occurring in the victim and the number of co-consumed drugs, low levels of methadone, under 600 ng/mL, could be expected in the post-mortem setting.

The post-mortem forensic evaluation of methadone-involving fatalities and the assessment of the causal value of methadone in death causation cannot disregard a multidisciplinary analysis, including the evaluation of any tolerance state and/or underlying diseases, which may have played a detrimental role on methadone acute toxicity.

## Figures and Tables

**Figure 1 metabolites-11-00189-f001:**
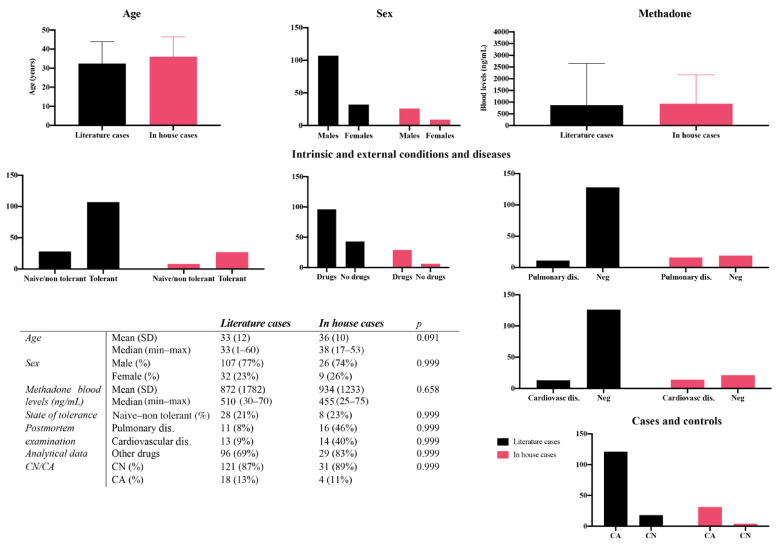
Epidemiological data and methadone blood concentrations of literature and in house cases, with results of t-tests (setting *p* < 0.05). SD: standard deviation. CA: cases; CN: controls. Dis: disease.

**Figure 2 metabolites-11-00189-f002:**
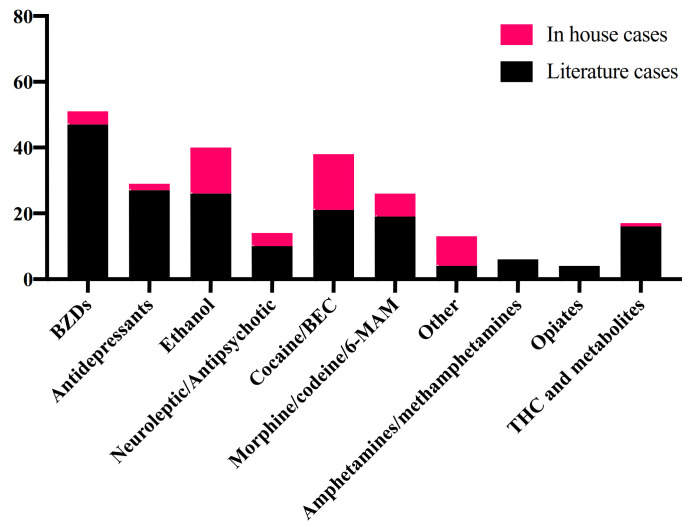
Other drugs detected in blood in literature and in-house cases. 6-MAM: 6-monoacetylmorphine; BEC: benzoylecgonine; BZDs: benzodiazepine; THC: tetrahydrocannabinol.

**Figure 3 metabolites-11-00189-f003:**
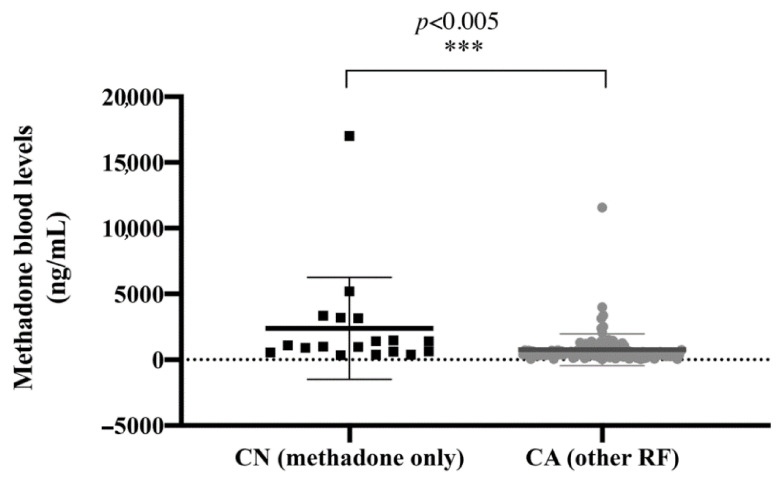
Comparison in blood methadone concentrations between controls (CN, i.e., cases in which methadone was the only cause of death, in black) and cases (CA, i.e., cases in which methadone was associated with other clinical-circumstantial (naive/non-tolerant state) or toxicological (other drugs detected in blood) conditions or with pulmonary or cardiovascular diseases, in red). Single values, as well as mean (thicker central line) and standard deviation (thinner lines) are shown. Two data outside the axis limit are not shown. *** = statistically significant result.

**Table 1 metabolites-11-00189-t001:** In-house cases, with extracted data and classification.

N	Age	Gender	Naive/ Non Tolerant	Post-Mortem Examination	Methadone (ng/mL)	6-MAM	Morphine	Codeine	Cocaine	BEG	BZDs	Ethanol	Other Drugs	Disease/ Condition
1	32	M	No	Past phlebitis, fresh injection marks L: 1560 g, edema, capillary congestion H: 420 g	48	neg	neg	neg	neg	pos	neg	pos	neg	D
2	42	M	No	Fresh injection marks L: 1800 g, edema, emphysema, capillary congestion H: 550 g	1040	neg	neg	neg	neg	neg	neg	neg	neg	-
3	52	M	No	Fresh injection marks L: 1490 g, edema, capillary congestion H: 420 g	1400	neg	neg	neg	neg	neg	neg	neg	neg	-
4	41	M	No	Fresh injection marks L: 1820 g, haemorrhagic edema, capillary congestion, anthracosis H: 460 g, focal myocardiosclerosis	550	neg	neg	neg	pos	pos	neg	neg	Promazine Lidocaine Phenacetin	D
5	31	M	Yes	Fresh injection mark L: 1480 g, haemorrhagic edema, capillary congestion H: 450 g, mild myocardiosclerosis	450	neg	pos	neg	neg	neg	neg	pos	neg	N, D
6	49	M	No	Past injection marks L: 1760 g, haemorrhagic edema, capillary congestion H: 420 g	791	neg	pos	neg	neg	neg	neg	neg	neg	D
7	34	M	No	Past phlebitis, fresh injection marks L: 1530 g, haemorrhagic edema, capillary congestion, peribronchial inflammation H: 500 g	433	neg	pos	neg	pos	pos	neg	neg	Phenacetine Paracetamol Levamisole Venlafaxine	D, P
8	40	M	Yes	Fresh injection mark L: 1410 g, haemorrhagic edema, capillary congestion, chronic inflammation H: 360 g, myocardiosclerosis, interstitial edema	32	neg	pos	neg	neg	neg	neg	neg	Paracetamol Theophylline	N, D, P, C
9	37	M	No	Past phlebitis L: 1400 g, haemorrhagic edema, capillary congestion, chronic inflammation H: 340 g, myocardiosclerosis, interstitial edema	338	neg	neg	neg	pos	pos	neg	neg	Ecgonine methylester	D, P, C
10	35	M	No	Fresh injection mark. L: 1690 g, hemorrhagic edema, capillary congestion, bronchopneumonia H: 460 g, edema, myocardiosclerosis, lipomatosis	1250	neg	neg	neg	neg	neg	neg	pos	Valproate Paracetamol	D, P, C
11	26	M	No	L: 1600 g, edema, bronchopneumonia H: 350 g	455	neg	neg	neg	neg	neg	neg	neg	Neg	P
12	41	M	No	Fresh injection marks L: 940 g haemorrhagic edema, capillary congestion, emphysema H: 320 g	495	neg	neg	neg	pos	pos	neg	neg	Ecgonine methylester	D, P
13	27	M	No	Multiple past injection marks L: 1505 g, haemorrhagic edema, capillary congestion H: 300 g, interstitial edema	270	pos	pos	pos	neg	neg	neg	neg	Neg	D
14	53	F	No	Past phlebitis L: 1270 g, haemorrhagic edema, capillary congestion H: 240 g mild coronarosclerosis	5200	neg	neg	neg	neg	neg	neg	pos	Neg	D
15	32	M	No	Fresh injection marks L: 1900 g, haemorrhagic edema, capillary congestion, emphysema, bronchopneumonia H: 440 g, mild coronarosclerosis, severe myocardiosclerosis	209	neg	neg	neg	pos	pos	neg	neg	Ecgonine methylester Levamisole Methylecgonine Cinnamate	D, P, C
16	25	F	No	L: 1400 g, edema, capillary congestion H: 246 g	3350	neg	neg	neg	neg	neg	neg	neg	neg	-
17	46	M	No	L: 1680 g, haemorrhagic edema, capillary congestion, bronchopneumonia H: 400 g, interstitial edema	517	neg	neg	neg	pos	pos	neg	pos	Ecgonine methylester Levamisole	D, P
18	44	M	No	Nasal mucosa atrophy L: 1980 g, haemorrhagic edema, bronchopneumonia H: 390 g, mild myocardiosclerosis, wavy myocardial fibers	122	neg	neg	neg	pos	pos	neg	pos	Ecgonine methylester Methotrimeprazine	D, P, C
19	26	M	No	L: 2010 g, haemorrhagic edema, bronchopneumonia H: 485 g myocarditis, pericarditis	158	neg	neg	neg	pos	pos	neg	pos	Neg	D, P, C
20	26	F	Yes	L: 1300 g, haemorrhagic edema, capillary congestion H: 265 g, myocardiosclerosis, wavy myocardial fibers	25	neg	neg	neg	pos	pos	neg	pos	Paracetamol	N, D, C
21	43	F	No	Fresh injection mark L: 1240 g, haemorrhagic edema, capillary congestion H: 450 g	3990	neg	pos	pos	neg	neg	neg	pos	Neg	D
22	44	M	No	Past phlebitis L: 1700 g, haemorrhagic edema H: 480 g, myocardiosclerosis, wavy myocardial fibers	1450	neg	pos	pos	pos	pos	neg	pos	Ecgonine methylester Fluconazole Levamisole Trimethoprim	D, C
23	38	M	No	Past phlebitis, skin necrosis L: 1240 g, haemorrhagic edema, capillary congestion H: 315 g, severe myocardiosclerosis	454	neg	neg	neg	neg	pos	neg	pos	neg	D, C
24	42	M	No	Past phlebitis, fresh injection marks L: 1630 g, haemorrhagic edema, capillary congestion H: 440 g, myocardiosclerosis	460	neg	neg	neg	pos	pos	neg	pos	neg	D, C
25	47	M	No	L: 1650 g, edema, bronchopneumonia H: 400 g, myocardiosclerosis, myocarditis, pericarditis	187	neg	neg	neg	neg	neg	Delorazepam Lorazepam	neg	Clorpromazine Clotiapine Clozapine	D, P, C
26	19	M	Yes	Fresh injection marks L: 2060 g, haemorrhagic edema, pneumonia H: 400 g, interstitial edema	77	neg	neg	neg	neg	neg	neg	neg	neg	N, P
27	18	M	No	L: 1260 g, edema, capillary congestion H: 380 g	3190	neg	neg	neg	neg	neg	neg	neg	neg	-
28	48	F	No	L: 1780 g, haemorrhagic edema, capillary congestion H: 320 g	2392	neg	neg	neg	neg	pos	neg	neg	neg	D
29	19	F	Yes	L: 640 g, haemorrhagic edema, interstitial bronchopneumonia H: 210 g	82	neg	neg	neg	neg	neg	Alprazolam	pos	neg	N, D, P
30	43	M	No	L: 1250 g, haemorrhagic edema, emphysema H: 540 g, mild myocardiosclerosis	716	neg	neg	neg	neg	neg	Alprazolam Diazepam Lorazepam Nordiazepam Oxazepam	pos	Mirtazapine	C, D
31	17	M	Yes	L: 1606 g, haemorrhagic edema, bronchopneumonia H: 324 g, septic emboli, wavy myocardial fibers	122	neg	neg	neg	pos	pos	neg	neg	neg	N, D, P, C
32	47	M	No	L: 2000 g, edema, peribronchial inflammation, emphysema H: 420 g	117	neg	neg	neg	neg	pos	neg	neg	THC-COOH	D, P
33	26	F	Yes	L: 960 g, haemorrhagic edema H: 310 g	1209	neg	neg	neg	pos	pos	neg	neg	Cocaethylene Ephedrine	N, D
34	28	F	Yes	L: 1080 g, edema, capillary congestion H: 290 g, interstitial edema	645	neg	neg	neg	pos	pos	neg	neg	neg	N, D
35	44	F	No	L: 1020 g, haemorrhagic edema, bronchopneumonia, emphysema, anthracosis H: 253 g, severe myocardiosclerosis	451	neg	neg	neg	neg	neg	Lorazepam	neg	neg	D, P, C

N: number of the case. Gender. F: female, M: male. H: heart; L: lung. neg: no drug detected in blood; pos: positive detection in blood. 6-MAM: 6-monoacetylmorphine; BEG: benzoylecgonine; BZDs: benzodiazepine; THC-COOH: tetrahydrocannabinol carboxylic acid. D: drugs; N: naive/non-tolerant; P: pulmonary; C: cardiac.

**Table 2 metabolites-11-00189-t002:** Results of the single linear regression applied for all independent variables with and without dependent variable transformations.

Methadone Levels		Variable *a* (Tolerance State)	Variable *b* (Pulmonary Diseases)	Variable *c* (Cardiovascular Diseases)	Variable *d* (Other Drugs)	Variable *e* (Number of Drugs)	Variable *f* (Number of Conditions/Diseases)
not modified	*p* value	0.1610	0.1282	0.1544	0.0934 *	0.0307 *	0.0031 *
log 10 transf	*p* value	0.0079 *	0.0289 *	0.1434	0.0022 *	0.0003 *	<0.0001 *
	equation	*y* = −0.1763 * *x* + 0.6795	*y* = −0.1294 * *x* + 0.5019	*y* = −0.08746 * *x* + 0.3896	*y* = −0.2234 * *x* + 1.323	*y* = −0.8232 * *x* + 3.724	*y* = −0.6587 * *x* + 3.018

log 10 transf: transformation of the dependent variable on a log10 basis. * = statistically significant results.

**Table 3 metabolites-11-00189-t003:** Results of the multiple linear regression analysis between methadone blood concentration, transformed on a logarithmic base, and independent variables.

Variables in the Equation	Standardized Coefficient β	Coefficients Standard Error	β Lower-Upper Limits	*p*
Variable *a*—tolerance state	−0.214	0.003	−1.038–−0.377	0.003 *
Variable *b*—pulmonary diseases	−0.175	0.036	−1.144–−0.397	0.036 *
Variable *c*—cardiovascular diseases	−0.082	0.322	−0.963–−0.043	0.322
Variable *d*—other drugs	−0.283	0.074	−1.063–−0.143	0.001 *

β: regression coefficient, estimated factor effects on log-methadone concentration (log-ng/mL). * = statistically significant results

**Table 4 metabolites-11-00189-t004:** Results of the binomial logistic regression analyses for variables *a*–*d* (I) and *e*–*f* (II).

Binomial Logistic Regression	Variables in the Equation	β	Standard Error	Exp (B) and 95% C.I.	*p*
I. Nagelkerke R^2^ = 0.15	Variable *a*—tolerance state	1.100	0.463	3.003 (1.213–7.437)	0.017 *
	Variable *b*—pulmonary diseases	1.551	0.618	4.717 (1.404–15.847)	0.012 *
	Variable *c*—cardiovascular diseases	0.327	0.568	0.721 (0.237–2.195)	0.565
	Variable *d*—other drugs	1.207	0.388	3.344 (1.563–7.154)	0.002 *
II. Nagelkerke R^2^ = 0.14	Variable *e*—number of drugs	0.252	0.126	1.286 (1.005–1.646)	0.045 *
	Variable *f*—number of conditions/diseases	0.736	0.247	2.088 (1.287–3.388)	0.003 *

β: regression coefficient. Exp (B): odds ratio. * = statistically significant results.

## Data Availability

The data presented in this study are available in Table 1 and Appendix A

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
