# Peer review of "The Role of Risk or Contributory Death Factors in Methadone-Related Fatalities: A Review and Pooled Analysis"

_metabolites, 2021, doi:10.3390/metabo11030189_

Round 1

Reviewer 1 Report

Dear Authors,

this study is comprehensively investigated and prepared as a manuscript and also presents data significant to the field. However, I have several minor or major suggestions regarding this paper:

  • Line 27 - the word 'naive' has a typo and should be written correctly

  • One of the limitations is that this systematic review has been performed with the usage of articles found only in one database (PubMed) which slightly decrease the reliability of the study. It would be recommended to check other databases as well and seek for other articles not included in PubMed but which might be crucial in this study. It would increase the reliability of this study.
  • The inclusion criteria as well as exclusion criteria (especially point number three) are very well described and indicate that this study was comprehensively investigated.
  • Line 227 - you have mentioned about 'Table A' in the Supplementary material, however as I see there is no such table there - could you indicate which one is that? Please change it for the proper name as used in the Supplementary material. I suppose that you were thinking about Table S1? Am I right?
  • Please check the whole manuscript because word 'naive' has a typo in it and it should be corrected.
  • Please check the whole manuscript once again in terms of English since I have detected several either minor or major mistakes which should be corrected before the manuscript can be processed further.

Best wishes with your further research!

Best regards

Author Response

Overall comment “Dear Authors, this study is comprehensively investigated and prepared as a manuscript and also presents data significant to the field. However, I have several minor or major suggestions regarding this paper:.

Point by point response

  1. “Line 27 - the word 'naive' has a typo and should be written correctly”

A: the error has been corrected across all manuscript by replacing and substituting the words “naïve” with “naive”.

  1. "One of the limitations is that this systematic review has been performed with the usage of articles found only in one database (PubMed) which slightly decrease the reliability of the study. It would be recommended to check other databases as well and seek for other articles not included in PubMed but which might be crucial in this study. It would increase the reliability of this study.”

A: we agree with the reviewer, this is certainly a drawback of the study which has now been listed in the “limitation” section. Even though other databases were not explored, we sought for other relevant articles by checking the references of the included studies, as shown in the Prisma flow chart at Figure S1 of the Supplementary material. Notwithstanding this, since we acknowledge that our study cannot be properly classified as a “systematic” review, we have modified the title accordingly.

  1. "The inclusion criteria as well as exclusion criteria (especially point number three) are very well described and indicate that this study was comprehensively investigated.”

A: we thank the reviewer for her/his appreciation.

  1. “Line 227 - you have mentioned about 'Table A' in the Supplementary material, however as I see there is no such table there - could you indicate which one is that? Please change it for the proper name as used in the Supplementary material. I suppose that you were thinking about Table S1? Am I right?”

A: we modified the names of figures and tables in the text and in the supplementary material accordingly.

  1. “Please check the whole manuscript because word 'naive' has a typo in it and it should be corrected”

A: the error has been corrected across all the manuscript by replacing and substituting the words “naïve” with “naive”.

  1. “Please check the whole manuscript once again in terms of English since I have detected several either minor or major mistakes which should be corrected before the manuscript can be processed further.”

A: thanking the reviewer for her/his suggestion, we have performed an in-depth English revision by checking grammar and spelling.

Reviewer 2 Report

Usage of a forward slash in the title (and passim) seemed wrong, so I consulted AMA stylebook - in short - it's better not to use it. 

The introduction to this paper provides a sane and informative background of the subject, instantly giving insight into its relevance and the wide range of post-mortem findings. I think that the motivations for this review should be made more explicit; the rationale for database selection for literature search should be defined unambiguously. This is my main reproach to this article - database selection for literature search whar implies flaws regarding “relevant criteria” establishment and overall search strategy. Assuming it is designed well, a pooled analysis is an excellent choice of literature reviews to combine the results of multiple epidemiological studies. However, pooled analyses can only be conducted if the included studies used the same study design and statistical models, and if their respective populations were homogeneous.

Author Response

Reviewer #2:

  1. “Usage of a forward slash in the title (and passim) seemed wrong, so I consulted AMA stylebook - in short - it's better not to use it.”

A: we corrected the title which is now: “The role of risk or contributory death factors in methadone-related fatalities: a review and pooled analysis”

  1. "The introduction to this paper provides a sane and informative background of the subject, instantly giving insight into its relevance and the wide range of post-mortem findings. I think that the motivations for this review should be made more explicit.”

A: we modified the introduction highlighting that there is a need to acquire more in depth knowledge regarding the risk or contributory factors for methadone deaths.

  1. "The rationale for database selection for literature search should be defined unambiguously. This is my main reproach to this article - database selection for literature search whar implies flaws regarding “relevant criteria” establishment and overall search strategy. Assuming it is designed well, a pooled analysis is an excellent choice of literature reviews to combine the results of multiple epidemiological studies. However, pooled analyses can only be conducted if the included studies used the same study design and statistical models, and if their respective populations were homogeneous.”

A: we agree with the reviewer that the homogeneity among included studies is a relevant issue affecting pooled analyses. Unfortunately, literature retrospective case series and articles regarding the topic are very varied, so that it would be impossible to include only studies with the same study design and statistical models. We tried our best to overcome this limitation, by reducing the number of included cases only to those articles which provided extractable data, so that we could reduce to the minimum the differences among studies. Moreover, in the revised version of the manuscript, we added this relevant drawback to the limitation section of the paper.

Reviewer 3 Report

The authors should improve the Conclusion part of the manuscript.

Author Response

  1. “The authors should improve the Conclusion part of the manuscript”

A: As suggested, we have improved the conclusions of the manuscript, hoping to have provided a clearer view of the background, methods and results of our paper.

Round 2

Reviewer 1 Report

Dear Authors,

thank you for correcting your manuscript according to my suggestions.

I have no further comments regarding this manuscript.

Best luck with your further research

Best regards

Reviewer